# On Surgical Fine-tuning for Language Encoders

**Abhilasha Lodha**[1]*, **Gayatri Belapurkar**[1]*, **Saloni Chalkapurkar**[1]*, **Yuanming Tao**[1]*,
**Reshmi Ghosh**[2], **Samyadeep Basu**[3], **Dmitrii Petrov**[1], **Soundararajan Srinivasan**[2]

[1]University of Massachusetts, Amherst
[2]Microsoft Corp.
[3]University of Maryland, College Park
Correspondence to: `reshmighosh@microsoft.com`

## Abstract

Fine-tuning all the layers of a pre-trained neural language encoder (either using all the parameters or using parameter-efficient methods) is often the de-facto way of adapting it to a new task. We show evidence that for different downstream language tasks, fine-tuning only a subset of layers is sufficient to obtain performance that is close to and often better than fine-tuning all the layers in the language encoder. We propose an efficient metric based on the diagonal of the Fisher information matrix (FIM score), to select the candidate layers for selective fine-tuning. We show, empirically on GLUE and SuperGLUE tasks and across distinct language encoders, that this metric can effectively select layers leading to a strong downstream performance. Our work highlights that task-specific information corresponding to a given downstream task is often localized within a few layers, and tuning only those is sufficient for strong performance[1]. Additionally, we demonstrate the robustness of the FIM score to rank layers in a manner that remains constant during the optimization process.

## 1 Introduction

Fine-tuning of language encoders is a crucial step towards applying natural language processing solutions to real-world challenges. It allows adaptation of knowledge on target distribution after generally training on source data, but requires curation of adequately sized labelled dataset to gain 'new' knowledge while retaining information obtained during the pre-training phase.

Although preserving knowledge of target distribution by tuning the entire model can yield impressive results, it can be expensive and may increase data volume requirements. Additionally,

fine-tuning all layers arbitrarily might risk overfitting or adversely affecting the generalization ability of the model during the transfer learning process. While, recently, the focus has shifted to development of parameter efficient approaches of fine-tuning large-language and language models (Liu et al. (2022), Lialin et al. (2023), Han et al. (2021)), these techniques still require development of an 'adapter' architecture relative to a target dataset. We therefore focus on developing a data-driven criteria to automatically identify and tune a smaller sub-set of layers using only $\approx 100$ target data samples.

In this paper, we propose a simple strategy to select layers for fine-tuning for real-world NLP tasks, leveraging the Fisher Information Matrix (FIM) score, which quantifies the impact of parameter changes on a model's prediction. We further demonstrate the effectiveness of FIM score on language encoder model(s) on practical NLP tasks from GLUE and SuperGLUE benchmark, by identifying a subset of layers that are most informative to adapt to the target data distribution. We find that fine-tuning parameters in identified subset of layers by FIM score, outperforms full model fine-tuning in some, and results in comparable performance to full fine-tuning in almost all the GLUE and Super-GLUE tasks. In niche scenarios, where FIM scores leads to selection of layers that contribute to suboptimal performance in comparison with full model fine-tuning, we investigate the nuanced characteristics of the GLUE and SuperGLUE tasks through the lens of linguistic features learned during transfer learning process, and potential categories of target data distribution shifts that could influence the performance while we surgically fine-tune.

Interestingly, we find that GLUE and Super-GLUE tasks that are dependent on a simpler understanding of linguistic features such as syntax and semantics as well as discourse understanding, can be surgically fine-tuned using our proposed

---

*Equal Contribution: order determined by the alphabetical arrangement of first names.

[1]Our code is publicly available at: Github

FIM score criteria. However, we find that for tasks that rely on learning more complex knowledge of both high and low level linguistic features such as textual entailment, common sense and world knowledge FIM score criteria unperformed to select the relevant layers. On investigating categories of target distribution shifts that could surface in various GLUE/SuperGLUE tasks, we also find that FIM score signals at efficient tuning of parameters for the group of tasks that align closely with concepts of domain, environmental, and demographic shifts, but fails to perform optimally on tasks that require learning of temporal drifts in language.

## 2 Related Work

Surgical fine-tuning has been widely explored in various computer vision applications to identify definitive distribution shifts in target datasets. Lee et al. (2023) explains why surgical fine-tuning could match or outperform full fine-tuning on distribution shifts and proposes methods to efficiently select layers for fine-tuning. However, in natural language applications, it is challenging to define such delineations due to the rapid changes in language based on context, domain, and individuals.

Several attempts have been made to conduct Parameter Efficient Fine Tuning (PEFT) in natural language. Lialin et al. (2023) and Han et al. (2021) explores the landscape of utilizing adapter layers and soft prompts, including additive methods, selective methods, and reparameterization-based techniques, whereas He et al. (2022) applies network pruning to develop a pruned adapter (sparse dapter). In another body of work, Sung et al. (2021) constructed a FISH (Fisher-Induced Sparse uncHanging) mask to choose parameters with the largest Fisher information. Additionally, Hu et al. (2021) attempted to efficiently fine-tune by proposing a Low Rank Adapter that reduces trainable parameters by freezing the weights of the pre-trained model and injecting trainable rank decomposition matrices into each layer of the architecture.

Some fine tuning techniques involve fine-tuning a small subset of the model parameters. Sung et al. (2022) propose a way of reducing memory requirement by introducing Ladder Side Tuning (LST). A small 'ladder side' network connected to each of the layers of the pre-trained model is trained to make predictions by taking intermediate activations as input from the layers via shortcut connections called ladders. Liu et al. (2022) demonstrated

the advantages of few-shot parameter-efficient fine-tuning over in-context learning in terms of effectiveness and cost-efficiency. Additionally, techniques like prompt tuning are also considered as parameter-efficient fine-tuning methods.

A handful of studies have investigated the knowledge gain in fine-tuning process for Language Encoders, particularly BERT. Merchant et al. (2020), Hessel and Schofield (2021) investigated the impact of shuffling the order of input tokens on the performance of the BERT model for several language understanding tasks. Sinha et al. (2021) further investigates the effectiveness of masked language modeling (MLM) pre-training and suggests that MLMs achieve high accuracy on downstream tasks primarily due to their ability to model distributional information.

However, our approach of efficient fine-tuning using the proposed FIM score criteria (that is able to capture signals from $\approx 100$ target data samples), differs from all existing methods, as it focuses on helping NLP practitioners with small size target datasets to efficiently rank and select important layers for optimizing the fine-tuning process.

## 3 Proposed Method

### 3.1 Fisher Information Matrix score

The significance of a parameter can be assessed by examining how modifications to the parameter affect the model's output. We denote the output distribution over $y$ generated by a model with a parameter vector $\theta \in \mathbb{R}^{|\theta|}$ given input $x$ as $p_\theta(y|x)$. To quantify the impact of parameter changes on a model's prediction, one approach is to compute the Fisher Information Matrix (FIM), which is represented by equation 1:

$$F_\theta = \mathbb{E}x \sim p(x) \left[ \mathbb{E}y \sim p_\theta(y \mid x) \nabla_\theta \log p_\theta(y \mid x) \nabla_\theta \log p_\theta(y \mid x)^\mathrm{T} \right] \quad (1)$$

where, $F_\theta$ is the FIM for the model with parameter vector $\theta$, quantifying the impact of parameter changes on the model's prediction, $\mathbb{E}x \sim p(x)$ is expectation operator over $x$ drawn from the distribution $p(x)$, $\mathbb{E}y \sim p_\theta(y \mid x)$ is the expectation operator over $y$ drawn from the output distribution $p_\theta(y \mid x)$, $\nabla_\theta$ is gradient operator with respect to the parameter vector $\theta$, $\log p_\theta(y \mid x)$ is the logarithm of the conditional probability of $y$ given $x$ under the model with parameter vector $\theta$, and $\nabla_\theta \log p_\theta(y \mid x) \nabla_\theta \log p_\theta(y \mid x)^\mathrm{T}$ is the outer product of the gradients, which is used to compute the FIM.

To analyze the impact of individual layers, we aggregate the diagonal elements of the FIM using the Frobenius norm. In our experimental setup, we randomly select a small sample (100 samples) from the validation set for each task. For fine-tuning, we specifically choose the top $5^2$ layers with the highest FIM scores. The FIM score measures the amount of information provided by an observable random variable about an unknown parameter in its distribution. It reflects the sensitivity of the likelihood function to changes in parameter values. A higher Fisher information score indicates that more information can be gleaned from the data regarding the parameter, leading to a more precise estimation of the parameter. In essence, a higher score suggests that the likelihood function is more responsive to changes in parameter values, improving the precision of parameter estimation.

## 4 Layer-wise Fisher Information Score Does Not Change During Fine-tuning

In Fig. (2), we compute the rank of distinct layers leveraging the Fisher Information Score across the fine-tuning process of BERT at different epochs. Across tasks including WSC and WIC, we find that the rank of the different layers remain more or less consistent across the entire optimization trajectory during fine-tuning. This shows that the layers which are important for a given task, can indeed be selected even before the start of fine-tuning and after pre-training is done. Using this observation, in the next section, we show the effectiveness of fine-tuning *only* the layers selected using Fisher Score at the start of the fine-tuning step.

## 5 Experiments and Results

### 5.1 Experimental Setup

We applied FIM score criteria to identify the 'layer importance rankings' for BERT[3] across real-world NLP tasks from GLUE and SuperGLUE (more details on experimental setup and hyperparameters in Appendix A.1). Based on these identified layer rankings, we performed surgical fine-tuning and

iteratively tuned parameters of top 1 to 5 most important layers, in the identified ranked layer order determined by FIM score, to compare and constrast the performance against full model fine-tuning.

Furthermore, to comprehensively understand scenarios where FIM scores leads to sub-optimal identification of layer ranks, we investigate the sensitivity of GLUE/SuperGLUE tasks (representing sentiment analysis, paraphrase detection datasets, NLI, question answering, linguistic acceptability, commonsense reasoning, etc.) with respect to four possible data distribution shift categories, namely:

**Domain shift:** Comparable shift from source data in target data distribution due to differences in fields or areas of knowledge.

**Environmental shift:** Changes from source data in target data distribution due to differences in contexts.

**Temporal drift:** Changes in use of certain language entities over time.

**Demographic shift:** Changes in data distribution across different demographic groups.

Additionally, we also qualitatively investigate influence of six primary linguistic features that are possibly influenced during the fine-tuning process depending on the task, namely Semantic Understanding, Discourse Understanding, Syntactic Understanding, Co-reference Resolution and Pronoun Disambiguation, Commonsense and World Knowledge, and Textual Entailment and Contradiction (for more details, refer to Appendix A.2).

### 5.2 Does surgical fine-tuning work across NLP tasks?

Our objective was to empirically analyze the performance of surgical fine-tuning approach leveraging FIM on real-world NLP tasks against full model fine-tuning. Results in Figure 1 (synthesized from Table 4 and Table 5) show that for GLUE and SuperGLUE tasks, surgical fine-tuning of identified most important layers results in comparable performance and sometimes outperforms tuning all parameters in all layers of BERT-base-cased model on target data distribution. Furthermore, we find that by selectively fine-tuning most relevant layer(s), as identified by FIM, the resulting performance on (*almost all*) GLUE and SuperGLUE tasks are in the ball park range of $\pm 5\%$ of the full fine-tuning performance.

We also discover that the identified layer impor-

---

[2]We select 5 layers from ranked importance of all layers in a language encoder, such as BERT-base as we find the surgical fine-tuning on at-most 5 layers across all GLUE & SuperGLUE tasks results, result in comparable performance when compared against full model fine-tuning

[3]We also validated the effectiveness of FIM with RoBERTa for some tasks to understand effectiveness of FIM scores across language encoder, results in Table ?? and Table ??

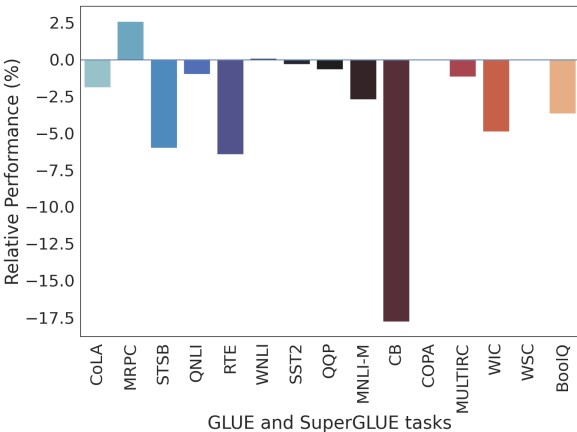

Figure 1: Plot of relative performance, i.e., the percentage point difference between the performance of surgical fine-tuning and full model fine-tuning, across GLUE and Super-GLUE tasks in two runs. Fine-tuning parameters in the ranked important layer(s) can outperform full fine-tuning, which is of significant importance, and in almost all tasks of GLUE and SuperGLUE, result in relative performance in the range of ±5%. Only for the case of RTE, CB, and COPA(showed no change) selected layers using FIM scores lead to sub-optimal results.

tance rank through FIM is different across settings, depending on the nature of task from GLUE and SuperGLUE benchmark.

## 5.3 Sensitivity of localized knowledge gain

For some tasks (RTE, STSB, CB, and COPA) FIM score based selected layers under-performed in surgical fine-tuning approach. Thus, we attempt to investigate through the lens of differences in learned linguistic features and possible distributional shifts in target data, the overall effectiveness of FIM for real-world NLP tasks.

### 5.3.1 Effect of linguistic features

Across the GLUE and SuperGLUE benchmarks, we observed that tasks requiring localized linguistic knowledge, such as discourse understanding (MNLI, MRPC, WNLI, WSC, and MultiRC), syntactic understanding (CoLA), semantic understanding, and commonsense/world knowledge (SST-2, QNLI), can be effectively fine-tuned with fewer localized parameters identified through ranked layer importance from the FIM scores.

However, for tasks that involve textual entailment (RTE) and require a strong grasp of common sense/world knowledge (COPA), as well as tasks focusing on understanding propositions (CB), surgically fine-tuning the model using FIM rankings resulted in sub-optimal performance. These tasks rely heavily on semantic understanding, logical rea-

soning, and the ability to integrate contextual information. Fine-tuning only a subset of layers based on FIM rankings may not adequately capture the necessary information and intricate relationships between linguistic elements, leading to decreased performance on these complex tasks.

We hypothesize that complex tasks such as RTE, COPA and CB, require a holistic understanding of language and reasoning abilities that span across multiple layers in the model. Consequently, selectively fine-tuning based solely on localized knowledge gain identified by FIM scores may not be sufficient to achieve optimal performance.

### 5.3.2 Effect of target data distributional shifts

We also investigate the effectiveness of FIM in suggesting the appropriate layer importance ranks that maximize the localization of knowledge while adapting to proposed categories of distributional shifts in target GLUE/SuperGLUE tasks.

When categorizing tasks based on their sensitivity to distribution shifts, it becomes evident that MNLI and MRPC tasks primarily revolve around the comprehension of semantic relationships within sentence pairs. These tasks exhibit a high sensitivity to shifts in the domain of discourse, as opposed to temporal or environmental variations. Conversely, tasks such as SST-2, CoLA, and QNLI heavily rely on contextual information to ascertain sentiment analysis accuracy, linguistic acceptability, and question answering, respectively. Consequently, these tasks are inclined towards being influenced by environmental shifts relative to the training data (for BERT-base) originating from Wikipedia and BookCorpus. Furthermore, STSB and RTE tasks demonstrate a notable level of change in the target data distribution with time as the language reference can change.

When comparing surgical fine-tuning with full fine-tuning in Figure 1, we observe that BoolQ and MRPC outperform the full model fine-tuning, while tasks such as QNLI, CoLA, MNLI, WSC, WiC, and MultiRC yield comparable performance. In contrast, RTE and STSB underperform in the surgical fine-tuning process. This indicates that our proposed approach of utilizing FIM to identify layer importance ranks works well in cases of domain and environmental shifts but fails to adapt to temporal drifts.

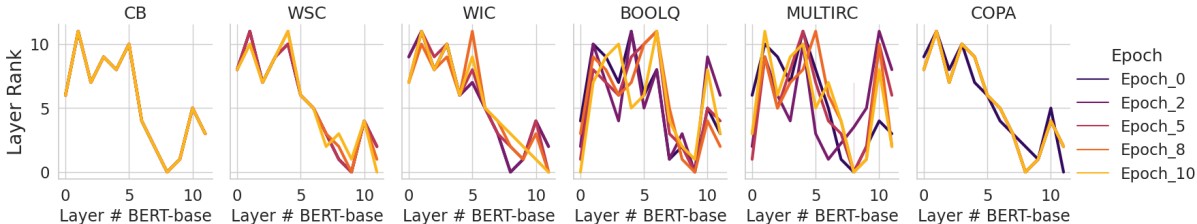

Figure 2: Plot of layer ranking for SuperGLUE tasks determined through FIM score during various checkpoints of optimization, i.e., epoch 0, 2, 5, 8, and 10. **The rank of layers with respect to Fisher Information remains constant across the optimization trajectory.** Note that for task CB, the layer rankings are identical across all the epochs.

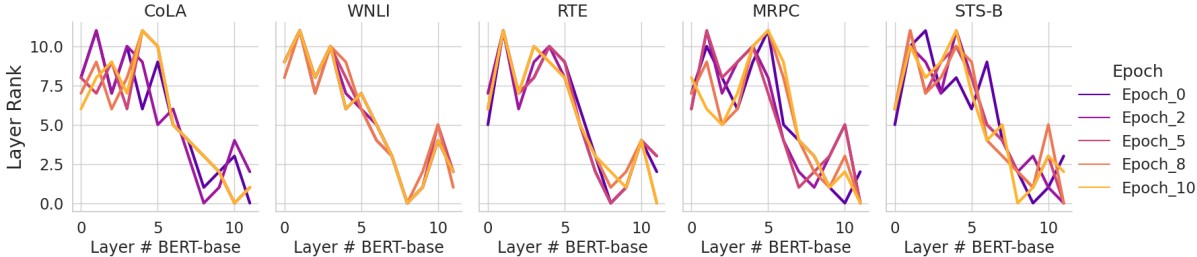

Figure 3: **Similar to SuperGLUE tasks, in Figure 2, we show that in GLUE tasks as well, the rank of layers with respect to Fisher Information remains constant across the optimization trajectory.** Especially for WNLI and RTE, the layer rankings are almost identical across the various checkpoints.

## 5.4 Ranking layers using FIM score vs. optimization trajectory

Upon investigating the efficacy of our proposed approach even further, we observed that the ranking of layers for surgical fine-tuning determined through FIM scores for SuperGLUE (Figure 2) and GLUE (Figure 3) tasks remains constant across various checkpoints of the optimization trajectory.

In particular, we investigate the rankings at epochs 0, 2, 5, 8, and 10 and observe that for Super-GLUE and GLUE tasks, the deviation in rankings is almost negligible (deviation plots in Appendix section A.3), and in some tasks like CB, WNLI, and RTE, the trajectory is identical. Thus, the arrangement of layers selected by the FIM score remains unchanged for the task at hand as the fine-tuning process progresses.

## 6 Conclusion

This paper contributes to the growing body of work that demonstrate that selective fine-tuning of language models is not only efficient but also effective in many downstream tasks. Summarizing our contributions, we strongly prove that selecting layers for finetuning based on ranking according to the FIM scores gives optimal results on a majority of GLUE and SuperGLUE tasks and could thus help NLP practitioners with small datasets to efficiently select a sub-set of relevant layers for optimized fine-

tuning for many real-world natural language tasks. In future work, we plan to investigate the linguistic correlates of different layers in large-language models (LLMs) and the value of FIM in surfacing them.

## Limitations and Future Work

The FIM score criteria proposed in this paper shows promising results on several GLUE and SuperGLUE tasks with language encoders. However, additional experiments are needed on some of the recent very large parameter models that perform well in zero-shot settings.

In addition, we plan to extend our evaluations and compare our method with existing solutions, such as Low-Rank Adaption (LoRA), to quantify the benefits of our approach.

## Acknowledgements

This work was supported by GPU resources from the University of Massachusetts, Amherst, and Microsoft Corporation, New England Research and Development Center.

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

# A Appendix

## A.1 Experiment Setup and Hyperparameters

We used our set of hyperparameters (mentioned in Table 1) to achieve close to State of the Art performance on almost all the GLUE and SuperGLUE tasks on fine-tuning the entire model. We then used this fully fine-tuned model as our baseline for comparing the performance after fine-tuning selected layers using FIM.

| Train Batch Size | 16 |
|---|---|
| Validation Batch Size | 16 |
| Learning Rate | 5e-5 |
| Epochs | 10 |
| Weight Decay | 0.01 |

Table 1: Hyperparameters used

We are using standard GLUE and SuperGLUE datasets and BERT (Devlin et al., 2019) as the transformer based language model for our experiments with FIM. For the datasets, we leveraged the HuggingFace Transformer library (Wolf et al., 2019) and used the full validation sets corresponding to the task to train the model and evaluate its performance (mentioned in Table 2 and 3). Code implementation is done using PyTorch and Python on Unity clusters (M40 and TitanX instances) and Google Colab.

## A.2 Task Specific Observations in BERT with Fisher metrics

We conducted various fine-tuning experiments on various GLUE and SuperGLUE tasks using BERT base cased model. The accuracy values obtained in these experiments are mentioned in Table 4 and 5.

1. **Selective Layer Fine-tuning:**

| QNLI | SST-2 | CoLA | MRPC | RTE | WNLI | STSB | QQP | MNLI-M |
|---|---|---|---|---|---|---|---|---|
| 5,463 | 872 | 1,043 | 408 | 277 | 71 | 1,500 | 40,430 | 9,815 |

Table 2: Validation Dataset size across GLUE tasks

| CB | COPA | MultiRC | WiC | WSC | BoolQ |
|---|---|---|---|---|---|
| 56 | 100 | 4,848 | 638 | 104 | 3,270 |

Table 3: Validation Dataset size across SuperGLUE tasks

(a) CoLA (Corpus of Linguistic Acceptability): CoLA focuses on grammatical acceptability, requiring a model to understand subtle linguistic nuances. It is likely that the lower layers captured in the Fisher metrics, which correspond to lower-level linguistic features, contain vital information for such tasks. By fine-tuning these layers, the model can still grasp the necessary linguistic patterns and achieve comparable performance.

(b) QQP (Quora Question Pairs) and QNLI (Question Natural Language Inference): These tasks involve semantic similarity and entailment recognition, which can be aided by lower-level features captured in the lower layers. Fine-tuning the top-5 layers from the Fisher metrics may still preserve enough relevant information to make accurate predictions.

(c) MultiRC (Multi-Sentence Reading Comprehension): MultiRC assesses comprehension across multiple sentences. It is possible that the lower layers encompass critical contextual information, enabling the model to capture relevant relationships and dependencies within the text. Fine-tuning these layers could provide sufficient context understanding for accurate predictions.

2. **Improved Accuracy with Fewer Layers:**

(a) MRPC (Microsoft Research Paraphrase Corpus): MRPC focuses on paraphrase identification, which often relies on local context and surface-level similarities. The top-3 layers selected from the Fisher metrics may effectively capture these aspects, resulting in improved performance compared to the full model.

(b) WNLI (Winograd Natural Language Inference): WNLI is a pronoun resolution task that requires understanding coreference. The consistent accuracies across all layers suggest that the task primarily relies on higher-level reasoning and discourse-level comprehension rather than fine-grained distinctions among layers. Thus, fine-tuning a smaller subset of layers may still achieve comparable performance.

(c) BoolQ (Boolean Questions): BoolQ involves answering boolean questions based on passages. As it requires logical reasoning and comprehension, the more streamlined approach of fine-tuning fewer layers (top layers given by Fisher) may enhance the model's ability to reason and derive accurate answers.

3. **Task-specific Variations:**

(a) WNLI and COPA: The tasks exhibit consistent accuracies across all layers, indicating that the distinctions among layers may not significantly impact the performance. WNLI's focus on pronoun resolution and COPA's requirement for causal reasoning might rely more on higher-level understanding and reasoning abilities rather than specific layer representations.

(b) WSC (Winograd Schema Challenge): WSC presents a challenging task that evaluates commonsense reasoning. Remarkably, the top accuracies, full model accuracy, and bottom-1 accuracies are all exactly the same. This result suggests that the layers identified by the Fisher Information Matrix capture the essential reasoning capabilities required for effective performance, underscoring the significance of higher-level features.

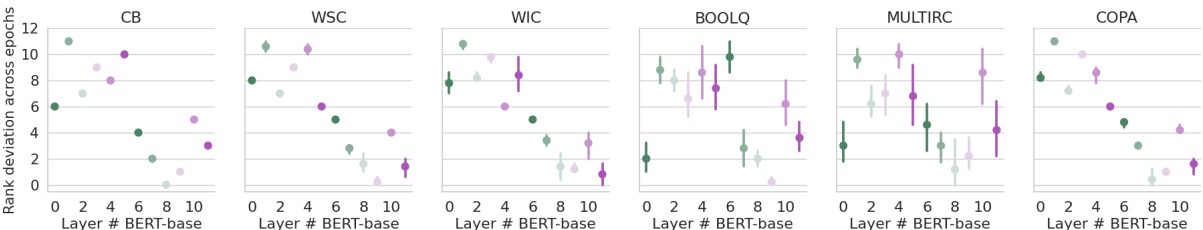

Figure 4: Degree of layer rank deviation (on y-axis) for SUPERGLUE tasks (complementary representation to Figure 2).

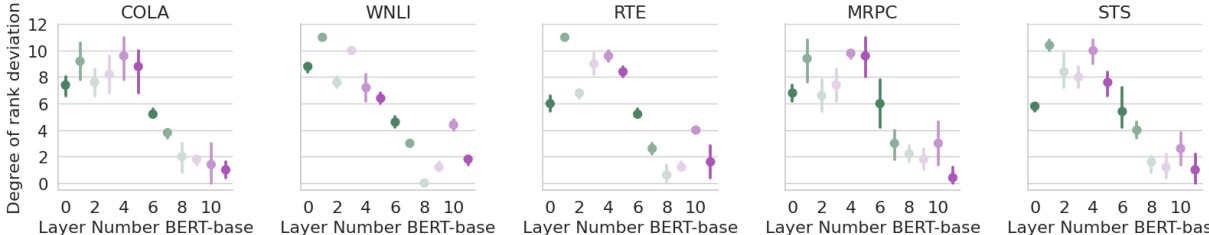

Figure 5: Degree of layer rank deviation (on y-axis) for GLUE tasks (complementary representation to Figure 3).

## A.3 FIM scores during optimization

Figures 4 and 5 are complementary extensions to Figure 2 and Figure 3, respectively, where we present evidence of minimal layer rank deviations (y-axis) for all SuperGLUE and GLUE tasks. For CB, WSC, COPA, WNLI, and RTE, the ranking of layers is identical for epochs 0, 2, 5. 8, and 10 of the training process.

## A.4 Results from BERT-base

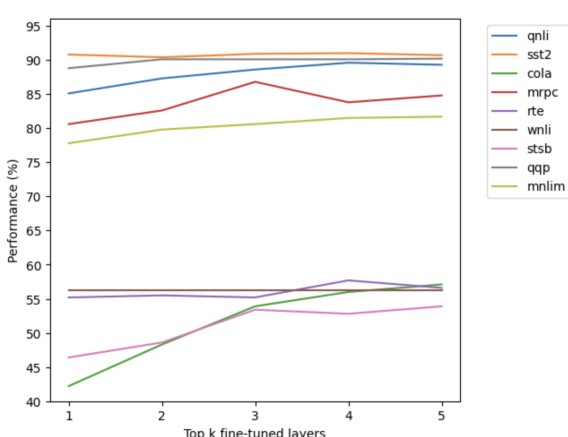

Figure 6: Performance of the GLUE Tasks when the top k layers identified by the fisher score are fine tuned

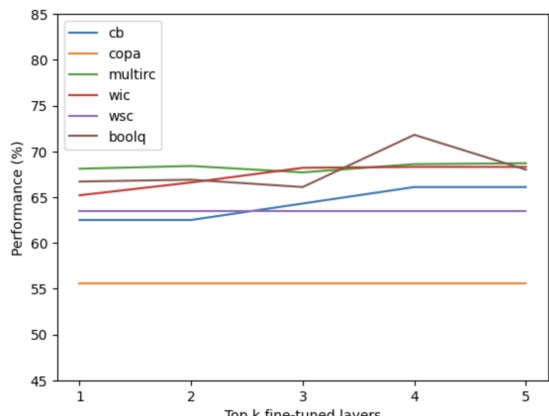

Figure 7: Performance of the SuperGLUE Tasks when the top k layers identified by the fisher score are fine tuned

| Layers finetuned | QNLI | SST-2 | CoLA | MRPC | RTE | WNLI | STSB | QQP | MNLI-M |
|---|---|---|---|---|---|---|---|---|---|
| State of the Art | 0.905 | 0.935 | 0.521 | 0.889 | 0.664 | 0.563 | 0.858 | 0.712 | 0.834 |
| Full-model | 0.905 | 0.923 | 0.582 | 0.846 | 0.620 | 0.563 | 0.574 | 0.907 | 0.829 |
| Top 1 | 0.851 | 0.908 | 0.422 | 0.806 | 0.552 | 0.563 | 0.464 | 0.888 | 0.778 |
| Top 2 | 0.873 | 0.904 | 0.483 | 0.826 | 0.555 | 0.563 | 0.486 | 0.901 | 0.798 |
| Top 3 | 0.886 | 0.909 | 0.539 | **0.868** | 0.552 | 0.563 | 0.534 | 0.901 | 0.806 |
| Top 4 | **0.896** | **0.910** | 0.560 | 0.838 | **0.577** | 0.563 | 0.528 | 0.901 | 0.815 |
| Top 5 | 0.893 | 0.907 | **0.571** | 0.848 | 0.566 | 0.563 | **0.539** | 0.902 | **0.817** |
| Bottom 1 | 0.899 | 0.885 | 0.208 | 0.684 | 0.581 | 0.563 | 0.525 | 0.903 | 0.730 |

Table 4: GLUE test results based on Fisher score on BERT base cased

| Layers finetuned | CB | COPA | MultiRC | WiC | WSC | BoolQ |
|---|---|---|---|---|---|---|
| State of the Art | 0.836 | 0.706 | 0.700 | 0.695 | 0.643 | 0.774 |
| Full-model | 0.804 | 0.556 | 0.695 | 0.718 | 0.635 | 0.707 |
| Top 1 | 0.625 | 0.556 | 0.681 | 0.652 | 0.635 | 0.667 |
| Top 2 | 0.625 | 0.556 | 0.684 | 0.666 | 0.635 | 0.669 |
| Top 3 | 0.643 | 0.556 | 0.677 | 0.682 | 0.635 | 0.661 |
| Top 4 | 0.661 | 0.556 | 0.686 | 0.683 | 0.635 | **0.718** |
| Top 5 | 0.661 | 0.556 | **0.687** | **0.683** | 0.635 | 0.680 |
| Bottom 1 | 0.393 | 0.556 | 0.658 | 0.636 | 0.635 | 0.681 |

Table 5: SuperGLUE test results based on Fisher score on BERT base cased