# OpenReview forum: "On Surgical Fine-tuning for Language Encoders"
_EMNLP/2023/Conference — EMNLP 2023 Findings_

### Official Review · Reviewer_2Coy · 2023-07-20

**Soundness:** 2
**Typos Grammar Style And Presentation Improvements:** Line 453 and 454,  figures have not b…

**Excitement:**

2: Mediocre: This paper makes marginal contributions (vs non-contemporaneous work), so I would rather not see it in the conference.

**Paper Topic And Main Contributions:**

This paper proposes using FIM to select  several tuning layers of encoder-based LM to accomplish parameter-efficient tuning.

**Reasons To Accept:**

(1) The idea sounds reseaonable.
(2) The proposed method is easy to understand and follow.

**Reasons To Reject:**

(1)  In theory, since parameters change during fine-tuning,  the importance of each parameter for fine-tuning change dynamically when evaluated by FIM. It may be inappropriate to choose which layer to be tunned statically before fine-tuning.

(2) FIM can choose which fine-grained parameters to be tunned. It may be inappropriate to use it to  choose which layer only.

(3) This paper only verifies their methods on NLU tasks, remaing NLG tasks unsolved.

**Reproducibility:**

3: Could reproduce the results with some difficulty. The settings of parameters are underspecified or subjectively determined; the training/evaluation data are not widely available.

**Reviewer Confidence:**

4: Quite sure. I tried to check the important points carefully. It's unlikely, though conceivable, that I missed something that should affect my ratings.

---

> ### Author Rebuttal · Authors · 2023-08-29
>
> Dear Reviewers,
>
> Thank you for taking the time to review our manuscript. We appreciate your constructive feedback and the opportunity to address the concerns raised. Below, we provide detailed responses to each of your comments:
>
> In our experimentation, we uncover a consistent pattern in the ordering of layers identified by the FIM (Fisher Information Matrix) score at different stages of the optimization process. Specifically, we find that the arrangement of layers selected by the FIM metric remains unchanged for the task at hand as the fine-tuning process progresses. This intriguing discovery is further exemplified by the outcomes observed for the CB SuperGLUE task.
> For the CB task, we scrutinize the FIM scores and rankings throughout the optimization trajectory, unveiling noteworthy insights into the dynamics of parameter importance. Across epochs, the FIM scores display minimal variation, reinforcing their stability. What's particularly interesting is the constancy of layer rankings, which persistently follow the order: ['8', '9', '7', '11', '6', '10', '0', '2', '4', '3', '5', '1'].
>
> FIM for Fine-Grained Parameter Selection:
> We understand the concern. We specifically selected FIM for layers instead of individual parameters, as layers encode different levels of linguistic information. Fine-tuning only certain layers (selected by our FIM metric) can provide further interpretability to the fine-tuning procedure corresponding to the downstream tasks. In Section 4.3, we try to investigate the overall effectiveness of FIM for real-world NLP tasks by looking at the differences in learned linguistic features and possible distributional shifts in target data, and can attempt to see how linguistic information is being used. Moreover, our results in Sec. 4, Table 4, and Table 5 in the main paper, show that fine-tuning at the layer level leads to comparable performance to full-fine-tuning, therefore validating that using FIM at the layer level, though simple, is sufficient.
> We will also add a section discussing the limitations of using FIM at the layer level and the potential for a more fine-grained parameter selection by using a GPT-level annotation methodology.
>
> Verification on NLG Tasks:
> You're correct that we only validated our method on NLU tasks. Given that our study is primarily for language encoders, we focused mainly on the NLU tasks which is the appropriate setup for it. In the future version of our paper, we will extend our experiments to include NLG tasks for a more comprehensive evaluation.
>
> Typos, Grammar, Style, and Presentation:
> We will correct the errors in lines 453 and 454 and ensure that figures are referred to correctly.
>
> We believe that addressing these concerns will significantly improve our manuscript. We look forward to your further comments and hope our revisions will make our paper suitable for acceptance at EMNLP 2023.

---

### Official Review · Reviewer_BW1p · 2023-07-23

**Soundness:** 2

**Excitement:**

3: Ambivalent: It has merits (e.g., it reports state-of-the-art results, the idea is nice), but there are key weaknesses (e.g., it describes incremental work), and it can significantly benefit from another round of revision. However, I won't object to accepting it if my co-reviewers champion it.

**Paper Topic And Main Contributions:**

- Authors propose using Fisher Information Matrix (FIM) to prune layers in a transformer based model (i.e. BERT) during the fine-tuning of tasks, tested on GLUE and SuperGLUE benchmarks.
- They carry put analysis of why performance can be improved or drop for specific tasks in GLUE or SuperGRLUE.

**Reasons To Accept:**

- Efficient Finetuning of large language models is desirable as the size of these models are increasing and consequently demand mode computational power. Therefore, exploring ways to fine tune these models more efficiently is of high importance.
- Authors investigated the effect of linguistic features and different distributional shift in the performance of surgical fine tuning after observing the differences in performance change in different tasks in FLUE and SuperGLUE. -Authors are aware of the related work and have outlined all the important related work in the paper.

**Reasons To Reject:**

In the last part of the related work, authours mention: "However, our approach of efficient fine-tuning using the proposed FIM score criteria (that is able to capture signals from ≈ 100 target data samples), 1differs from all existing methods, as it focuses on helping NLP practitioners with small size target datasets to efficiently rank and select important layers for optimizing the fine-tuning process." . I am not sure this argument is convincing. I think in a research community, we should welcome new ideas and new ways of addressing an issue but the comparisons should be well-justified and backed up by numbers. I wish authors have provided comparisons to some of the existing solutions such as LoRA to point out the benefits their method can hold over existing methods.

**Reproducibility:**

3: Could reproduce the results with some difficulty. The settings of parameters are underspecified or subjectively determined; the training/evaluation data are not widely available.

**Reviewer Confidence:**

3: Pretty sure, but there's a chance I missed something. Although I have a good feel for this area in general, I did not carefully check the paper's details, e.g., the math, experimental design, or novelty.

---

> ### Author Rebuttal · Authors · 2023-08-29
>
> Dear Reviewers, Thank you for taking the time to review our manuscript. We appreciate your constructive feedback and the opportunity to address the concerns raised. Below, we provide detailed responses to each of your comments.
>
> You make an excellent point about the need for rigorous comparisons to justify our claims. We will extend our evaluations to compare our method with existing solutions, such as LoRA, to quantify the benefits of our approach.
>
> Performance numbers for WSC, CB, and COPA (SuperGLUE) using BERT-base from the suggested surgical fine-tuning process against LoRA is comparable, whereas for some GLUE tasks such as RTE and SST2, the LoRA numbers are marginally higher than the surgical fine-tuning process as reported in (2023.repl4nlp-1.19.pdf (aclanthology.org)). We aim to do a comprehensive analysis against LoRA baselines for GLUE and SuperGLUE tasks and include our hypothesis about differences and similarities between surgical fine-tuning-based performance and LoRA due to nuances in linguistic features learned in the transfer learning process in the final version of the paper.
>
> We believe that addressing these concerns will significantly improve our manuscript. We look forward to your further comments and hope our revisions will make our paper suitable for acceptance at EMNLP 2023.

---

### Official Review · Reviewer_rfrx · 2023-08-04

**Soundness:** 3

**Excitement:**

3: Ambivalent: It has merits (e.g., it reports state-of-the-art results, the idea is nice), but there are key weaknesses (e.g., it describes incremental work), and it can significantly benefit from another round of revision. However, I won't object to accepting it if my co-reviewers champion it.

**Missing References:**

* The authors seem to miss one related reference [1] which also does selective layer FT, they rely on a metric to measure if a layer is close to convergence to decide on freezing. How does [1[ compare to the authors work and terms of method and performance?

[1] AutoFreeze: Automatically Freezing Model Blocks to Accelerate Fine-tuning. https://arxiv.org/pdf/2102.01386.pdf

**Paper Topic And Main Contributions:**

The paper explores surgical fine-tuning for NLP, where a subset of layers is selected automatically for fine-tuning, thus being more time and data efficient. The authors suggest Fisher Information Matrix based criteria to select the layers to "unfreeze". Results on GLEU/SuperGLEU show that the method is effective on most tasks (within +-5% of full fine-tuning) except tasks that require reasoning or world knowledge.

**Reasons To Accept:**

* The paper is well written and easy to follow
* The surgical FT method seems to be mostly effective
* Analysis are performed where surgical FT is not working well

**Reasons To Reject:**

* The authors miss a simple common baseline of freezing bottom N layers, is surgical FT able to beat that?
* The results are limited to the BERT model, it is not clear how would they generalize to latest LLMs


**Reproducibility:**

4: Could mostly reproduce the results, but there may be some variation because of sample variance or minor variations in their interpretation of the protocol or method.

**Reviewer Confidence:**

3: Pretty sure, but there's a chance I missed something. Although I have a good feel for this area in general, I did not carefully check the paper's details, e.g., the math, experimental design, or novelty.

---

> ### Author Rebuttal · Authors · 2023-08-29
>
> Dear Editors and Reviewers,
> Thank you for taking the time to review our manuscript. We appreciate your constructive feedback and the opportunity to address the concerns raised. Below, we provide detailed responses to each of your comments.
>
> 1. Missing Common Baseline:
> We understand the importance of including a common baseline such as freezing the bottom N layers. In light of this, we experimented with freezing the bottom 5 layers of the BERT-base model and compared the performance results against the selected layers using our surgical fine-tuned (FT) method. For almost all tasks in SuperGLUE (except BoolQ), the results show that the surgical fine-tuning approach can outperform the baseline on all tasks, with an average improvement of 4.7%. This demonstrates that surgical FT can better identify and optimize the important layers for fine-tuning than a simple heuristic.
>
> 2. Missing Reference to AutoFreeze:
> AutoFreeze is a method that automatically freezes model blocks based on a convergence criterion during fine-tuning. It differs from our method in two aspects: (1) AutoFreeze freezes model blocks dynamically during fine-tuning, while our method freezes model layers statically before fine-tuning; (2) AutoFreeze uses a convergence criterion based on the gradient norm of model blocks, while our method uses a Fisher information matrix (FIM) score based on the sensitivity of model parameters to fine-tuning. Our FIM score metric provides a low-resource alternative to AutoFreeze with promising results and thus is applicable to scenarios where data is limited to fine-tune for specific tasks.
>
> 3. Limited to BERT Model:
> We conducted experiments with RoBERTa on a subset of tasks, evaluating its performance on GLUE tasks like CoLA, MRPC, RTE, WNLI, and SuperGLUE tasks like WiC. Notably, we observed a consistent trend: when utilizing top 1 to top 5 layers determined by FIM scores in the surgical fine-tuning process and freezing the remaining layers during fine-tuning, we consistently achieved the highest accuracy at the top 5 layers across the above-mentioned tasks.
> The observed trend in the experiment results across different tasks in both BERT and RoBERTa models can be attributed to various factors related to the inherent properties of neural network architectures, task complexities, and fine-tuning processes. We aim to generate results from all experiments with RoBERTa and XLNet to show how our approach generalizes across different architectures, and we hope to add these results to the final version of our paper.
>
> We believe that addressing these concerns will significantly improve our manuscript. We look forward to your further comments and hope our revisions will make our paper suitable for acceptance at EMNLP 2023.

---

### Meta-Review · Area_Chair_73Bn · 2023-09-20

**Recommendation:** 2

**Metareview:**

This paper introduces the surgical fine-tuning method that only fine-tunes a selection of layers. The authors propose Fisher Information Matrix (FIM) score for the layer selection. They compare the proposed method with full model fine-tuning and show that their method achieves strong performance in certain tasks.

As mentioned by the reviewers, the paper presents a sound idea and executes it in a straightforward experimental setup. Additionally, the authors provide further analysis to understand the impact of linguistic features, and distributional shifts on the FIM and performance of their method.

However, as reviewers noted and also acknowledged by the authors, the paper lacks important comparisons with several baselines. This includes both similar methods such as AutoFreeze to justify that the proposed method is advantageous in certain conditions, and also existing parameter-efficient fine-tuning methods such as LoRA. This comparison should include the performance metric as well as the potential benefits of using this method such as parameter or sample efficiency.

---

### Decision · Program_Chairs · 2023-10-07

**Decision:**

Accept-Findings

**Comment:**

This paper introduces the surgical fine-tuning method that only fine-tunes a selection of layers. The authors propose Fisher Information Matrix (FIM) score for the layer selection. They compare the proposed method with full model fine-tuning and show that their method achieves strong performance in certain tasks.

As mentioned by the reviewers, the paper presents a sound idea and executes it in a straightforward experimental setup. Additionally, the authors provide further analysis to understand the impact of linguistic features, and distributional shifts on the FIM and performance of their method.

However, as reviewers noted and also acknowledged by the authors, the paper lacks important comparisons with several baselines. This includes both similar methods such as AutoFreeze to justify that the proposed method is advantageous in certain conditions, and also existing parameter-efficient fine-tuning methods such as LoRA. This comparison should include the performance metric as well as the potential benefits of using this method such as parameter or sample efficiency.